# The *MAOA* rs979605 Genetic Polymorphism Is Differentially Associated with Clinical Improvement Following Antidepressant Treatment between Male and Female Depressed Patients

**DOI:** 10.3390/ijms24010497

**Published:** 2022-12-28

**Authors:** Kenneth Chappell, Romain Colle, Jérôme Bouligand, Séverine Trabado, Bruno Fève, Laurent Becquemont, Emmanuelle Corruble, Céline Verstuyft

**Affiliations:** 1CESP, MOODS Team, INSERM UMR 1018, Faculté de Médecine, Université Paris-Saclay, F-94275 Le Kremlin Bicêtre, France; 2Service Hospitalo-Universitaire de Psychiatrie de Bicêtre, Hôpitaux Universitaires Paris-Saclay, Assistance Publique-Hôpitaux de Paris, Hôpital de Bicêtre, F-94275 Le Kremlin Bicêtre, France; 3Service de Génétique Moléculaire, Pharmacogénétique et Hormonologie de Bicêtre, Hôpitaux Universitaires Paris-Saclay, Assistance Publique-Hôpitaux de Paris, Hôpital de Bicêtre, F-94275 Le Kremlin Bicêtre, France; 4Plateforme d’Expertises Maladies Rares Paris-Saclay, Assistance Publique-Hôpitaux de Paris (AP-HP), Hôpitaux Universitaires Paris-Saclay, Assistance Publique-Hôpitaux de Paris, Hôpital de Bicêtre, F-94275 Le Kremlin Bicêtre, France; 5Unité Inserm UMRS 1185, Physiologie et Physiopathologie Endocriniennes, Faculté de Médecine, Université Paris-Saclay, F-94276 Le Kremlin-Bicêtre, France; 6Centre de Recherche Saint-Antoine, Institut Hospitalo-Universitaire ICAN, Service d’Endocrinologie, CRMR PRISIS, Sorbonne Université-INSERM, Hôpital Saint-Antoine, Assistance Publique-Hôpitaux de Paris, F-75012 Paris, France; 7Centre de Recherche Clinique, Hôpitaux Universitaires Paris-Saclay, Assistance Publique-Hôpitaux de Paris, Hôpital de Bicêtre, F-94275 Le Kremlin Bicêtre, France; 8Centre de Ressources Biologiques Paris-Saclay, Hôpitaux Universitaires Paris-Saclay, Assistance Publique-Hôpitaux de Paris, Hôpital de Bicêtre, F-94275 Le Kremlin Bicêtre, France

**Keywords:** pharmacogenetics, major depressive disorder, serotonin, MAO, antidepressant drugs

## Abstract

Major depressive disorder (MDD) is the leading cause of disability worldwide. Treatment with antidepressant drugs (ATD), which target monoamine neurotransmitters including serotonin (5HT), are only modestly effective. Monoamine oxidase (MAO) metabolizes 5HT to 5-hydroxy indoleacetic acid (5HIAA). Genetic variants in the X-chromosome-linked MAO-encoding genes, *MAOA* and *MAOB*, have been associated with clinical improvement following ATD treatment in depressed patients. Our aim was to analyze the association of *MAOA* and *MAOB* genetic variants with (1) clinical improvement and (2) the plasma 5HIAA/5HT ratio in 6-month ATD-treated depressed individuals. Clinical (*n* = 378) and metabolite (*n* = 148) data were obtained at baseline and up to 6 months after beginning ATD treatment (M6) in patients of METADAP. Mixed-effects models were used to assess the association of variants with the Hamilton Depression Rating Scale (HDRS) score, response and remission rates, and the plasma 5HIAA/5HT ratio. Variant × sex interactions and dominance terms were included to control for X-chromosome-linked factors. The *MAOA* rs979605 and *MAOB* rs1799836 polymorphisms were analyzed. The sex × rs979605 interaction was significantly associated with the HDRS score (*p* = 0.012). At M6, A allele-carrying males had a lower HDRS score (*n* = 24, 10.9 ± 1.61) compared to AA homozygous females (*n* = 14, 18.1 ± 1.87; *p* = 0.0067). The rs1799836 polymorphism was significantly associated with the plasma 5HIAA/5HT ratio (*p* = 0.018). Overall, CC/C females/males had a lower ratio (*n* = 44, 2.18 ± 0.28) compared to TT/T females/males (*n* = 60, 2.79 ± 0.27; *p* = 0.047). The *MAOA* rs979605 polymorphism, associated with the HDRS score in a sex-dependent manner, could be a useful biomarker for the response to ATD treatment.

## 1. Introduction

Major depressive disorder (MDD) is the current leading cause of disability worldwide [1]. Differences between sexes exist, too, as adult females are twice as likely to develop MDD and have a greater risk of experiencing longer and more severe depressive episodes compared to men [2]. Antidepressant drugs (ATD), which modulate monoamine neurotransmitter levels, remain a common treatment option for MDD. However, only about one-third of ATD-treated individuals achieve remission [3]. Identifying biomarkers to predict treatment response is thus an important clinical challenge.

Monoamine oxidase (MAO) metabolizes monoamine neurotransmitters, including serotonin (5HT) [4]. Two MAO proteins have been characterized, namely MAOA and MAOB, encoded by the neighboring X chromosome (Xp11.23) genes, *MAOA* and *MAOB*, respectively [4]. MAOA preferentially metabolizes 5HT and noradrenaline, while MAOB preferentially metabolizes phenylethylene and benzylamine [4,5]. MAO expression changes with age [4] and differs between males and females [5]. Smoking also decreases MAO activity [5] although former smokers exhibit increased platelet MAO activity [6]. MAOB is more abundant in the brain, and *MAOB* expression occurs mainly in serotonergic and histaminergic neurons and astrocytes, while *MAOA* is expressed in dopaminergic and noradrenergic neurons [4,7]. In the periphery, *MAOA* is predominantly expressed in fibroblasts, while MAOB is the sole MAO in platelets [4].

MAOA and MAOB density in various brain regions was increased in the context of a medication-free major depressive episode (MDE) compared to healthy controls [8,9]. MAOA density remained elevated in MDD individuals following treatment with a selective serotonin reuptake inhibitor (SSRI), although it was lower in individuals achieving remission compared to recurrent MDE sufferers [10]. Increased levels of plasma 5-hydroxyindoleacetic acid (5HIAA)—the main metabolite of 5HT—were also observed in unmedicated depressed individuals compared to healthy controls, suggesting greater 5HT metabolism in the context of MDD [11]. Since direct evaluation of MAO activity in the human brain is difficult, peripheral 5HT and 5HIAA levels may serve as a viable proxy [12]. Given their role in monoamine metabolism, MAOA and MAOB are interesting targets in neurological disorders [4,13]. As such, factors influencing their expression or function are of similar interest.

Genetic variants within the MAO-encoding genes have been studied in various psychiatric contexts, including depression, bipolar disorder, and schizophrenia [4,14]. Studies examining ATD response in MDD have focused notably on the upstream variable number tandem repeat (uVNTR) within the *MAOA* promoter. While some report no association with response [15,16,17,18], others report that the short-form uVNTR is associated with improved clinical outcomes [19,20,21]. Regarding *MAOA* polymorphisms, the rs6323 (NM_000240.4:c.891G>T) synonymous polymorphism has been the most analyzed. In several studies, the rs6323(T) allele was associated with improved treatment response [16,22,23] although in Han Chinese females with depression, it was closely associated with poorer response to venlafaxine [24]. The *MAOA* rs979605 (NM_000240.4:c.1262+69A>T/G) intronic polymorphism and rs6323 were also associated with reduced MAO enzymatic activity [5,25]. Regarding *MAOB*, it was observed that female TT homozygotes for the rs1799836 (NM_000898.5:c1348-36A>G) intronic polymorphism responded better to treatment [26]. Lower MAO activity was also observed in Swedish rs1799836(T) allele carriers [27].

Different factors complicate the analysis of X-chromosomal variants. For one, females carry two gene copies (and thus two alleles), while males carry one. However, X-chromosome inactivation (XCI) silences 1 of the gene copies in females. Still, factors such as XCI skewness (i.e., imbalanced allelic expression) and escape from XCI (i.e., expression from both silenced and active chromosomes) [28]—for which conflicting findings for *MAOA*/*MAOB* have been reported [29,30,31]—may influence *MAOA*/*MAOB* expression. Due to this complexity, X- and Y-linked variants are sometimes omitted from genome-wide association studies of ATD response [32,33]. Those that include them [34,35], however, may not adjust for these factors. As such, associations between X-chromosomal variants and clinical improvement after ATD treatment would benefit from controlling for these factors.

The associations of *MAOA*/*MAOB* genetic variants with clinical improvement after ATD treatment and MAO activity, which modulates monoamine concentrations—common targets of ATD therapy—qualify them as potential biomarkers for clinical improvement following ATD treatment in MDD. Given their X-chromosomal location, improved analysis of these genetic factors may help to explain sex differences in the context of MDD and its treatment. We attempted such an analysis by examining the association of *MAOA*/*MAOB* genetic variants with (1) clinical improvement following ATD treatment and (2) the plasma 5HIAA/5HT ratio as an estimate of MAO activity in a cohort of 6-month ATD-treated individuals with a current MDE in the context of MDD.

## 2. Results

### 2.1. Patient Demographics

Sociodemographic characteristics for the whole clinical and metabolomic cohorts and with respect to sex are shown in Table 1. For males (*n* = 119) and females (*n* = 259) of the clinical cohort, respectively, the mean age was 45.9 and 45 years, and 96% and 88% of subjects were Caucasian. Missing at follow-up rates relative to baseline were 4% and 6% after 1 month of ATD treatment (M1), 28% and 33% after 3 months (M3), and 46% and 47% after 6 months (M6).

### 2.2. Genetic Variant Selection

A total of 29 *MAOA* (26 single-nucleotide polymorphisms (SNP) and 3 insertion/deletions) and 25 *MAOB* (25 SNPs) genetic variants were identified. Among these, five *MAOA* SNPs and one *MAOB* SNP had a call rate ≥ 95% and a minor allele frequency (MAF) ≥ 5% (see Table 2). None significantly deviated from Hardy–Weinberg equilibrium (HWE). The five *MAOA* SNPs were in linkage disequilibrium (LD) (see Appendix A). The *MAOA* rs979605(A > G) genetic polymorphism was selected as a *MAOA* haplotype proxy for further analysis. The rs979605(G) allele was correlated with the rs6323(T) allele. Allelic frequencies did not significantly differ between males and females.

Sociodemographic characteristics with respect to sex and *MAOA* rs979605 and *MAOB* rs1799836 genotypes are shown in Table 3 and Table 4, respectively. Significant differences in the missing at follow-up rate (M1) and in age and the prescribed therapy were observed between female *MAOA* rs979605 genotypes of the clinical and metabolomic cohorts, respectively (see Table 3). Likewise, significant differences in ethnicity and in smoking status were observed between female *MAOB* rs1799836 genotypes of the clinical and metabolomic cohorts, respectively (see Table 4).

### 2.3. Clinical Improvement According to Variant Genotypes

We first examined associations of the *MAOA* rs979605 and *MAOB* rs1799836 genetic polymorphisms with clinical measures using the proposed full model for analyzing X-chromosomal variants, including the dominance term and polymorphism × sex interaction. In the model of the 17-item Hamilton Depression Rating Scale (HDRS) score, the *MAOA* rs979605 genetic polymorphism × sex interaction was significant after controlling for other factors (*F*_1,367_ = 6.32, *p* = 0.012), corresponding to a difference in the association of rs979605 with the HDRS score depending on sex (see Table 5). The *MAOB* rs1799836 genetic polymorphism was not a significant factor in models of clinical outcomes at a Bonferroni-corrected level of *p* < 0.025, and neither were its interactions with time or sex (see Table 5).

We further investigated the association between rs979605 and the HDRS score in A and G allele subgroups under different XCI status assumptions. When modeling the HDRS score in A allele carriers (*n* = 183) assuming random XCI (i.e., males coded the same as homozygous females), sex was a significant factor at a Bonferroni-corrected level of *p* < 0.0125 to account for the four models examined after controlling for other factors (*F*_1,187_ = 7.66, *p* = 0.0062), corresponding to a difference in the association of the *MAOA* rs979605 A allele with the HDRS score depending on sex (see Appendix A). At M6, a significantly higher HDRS score was observed in female rs979605 AA homozygotes (*n* = 14, 18.1 ± 1.87) compared to male A carriers (*n* = 24, 10.9 ± 1.61) following Bonferroni correction (coef. = 7.21, 95%CI [3.03–11.38], *p* = 0.0067) (see Figure 1). No other allele-specific associations with the HDRS score were observed.

We also examined the association of the *MAOA* rs979605 genetic polymorphism in male and female subgroups. However, neither it nor its interaction with time was a significant factor in either male or female populations after controlling for other factors.

### 2.4. MAO Activity Estimation According to Variant Genotypes

We also examined the associations of rs979605 and rs1799836 with the plasma 5HIAA/5HT ratio using the proposed full model. The plasma 5HIAA/5HT ratio was log_2_ transformed due to non-normality (see Appendix A). Baseline plasma 5HIAA/5HT ratios (mean = 8.84, SD = 11.15) were lower compared to ratios at M3 (mean = 15.02, SD = 12.75) and M6 (mean = 15.34, SD = 16.75). After controlling for other factors, including significant ones (i.e., age, ATD class, ATD-naïve status, and time), the plasma 5HIAA/5HT ratio was significantly associated with the *MAOB* rs1799836 genetic polymorphism (*F*_1,133_ = 5.92, *p* = 0.016) (see Table 6). In a model without the non-significant sex interactions, rs1799836 remained significant after controlling for the same factors (*F*_1,135_ = 6.00, *p* = 0.016) (see Appendix A). Globally, TT/T females/males had a significantly higher 5HIAA/5HT ratio (2.79 ± 0.27) compared to CC/C females/males (2.18 ± 0.28) following Bonferroni correction for the three comparisons (coef. = 0.61, 95%CI [0.12–1.11], *p* = 0.047) (see Figure 2).

We also examined whether baseline or serial log_2_-transformed plasma 5HIAA/5HT ratios were associated with clinical outcomes. However, neither 5HIAA/5HT ratios nor their interactions with rs979605 and rs1799836 genotypes or time were significant factors in models of clinical outcomes after controlling for other factors.

### 2.5. Power Analysis

We had sufficient power to detect an *R*^2^ = 3% for the HDRS score (SD = 7.48) between genotypes for both rs979605 and rs1799836 among the 259 individuals at M3 and an *R*^2^ = 5.2% for baseline plasma 5HIAA/5HT ratio (SD = 1.97) variation between genotypes for both polymorphisms among the 148 individuals of the metabolomic cohort.

## 3. Discussion

In this ancillary investigation of METADAP, we analyzed the association between the rs979605 and rs1799836 genetic polymorphisms of *MAOA* and *MAOB*—encoding for the monoamine oxidases—respectively, and clinical improvement following ATD treatment in depressed patients. The allele frequency of rs1799836(C) in METADAP (45.1%) was comparable to that of non-Finnish Europeans of gnomAD (45.6%) [36], but rs979605(A) was more frequent in METADAP (34.8%), notably in males (38.2%) compared to males of this same gnomAD population (29.7%). The *MAOB* rs1799836 genetic polymorphism was not significantly associated, and neither were its interactions with time or sex, with clinical outcomes. The rs979605 × sex interaction was significantly associated with the HDRS score, and after 6 months of treatment, a significantly higher HDRS score was observed in female rs979605 AA homozygotes compared to male A carriers.

The *MAOA* rs6323 polymorphism—in LD with rs979605—was previously examined in association with response to ATD treatment in several contexts, including rapid versus delayed response and response in placebo- versus ATD-treated individuals [16,22]. Its analysis in two cohorts of depressed patients (baseline HDRS score ≥18) following several weeks of ATD treatment corresponds best with our own [23,24]. However, where Bi et al. observed that rs6323(T) was associated with poorer clinical outcomes in 6-week venlafaxine-treated Han Chinese females [24], Tadić et al. observed that, among females of their European population, TT homozygotes had significantly improved HDRS scores compared to G allele carriers after 4 weeks of mirtazapine treatment [23]. The former finding may be due to ethnic differences. Indeed, in our analysis of predominantly Caucasian (~90%) and 6-month SSRI- or SNRI-treated (81%) depressed patients (baseline HDRS score ≥18), we observed that females homozygous for rs979605(G)—in LD with rs6323(T)—had decreased HDRS scores compared to AA homozygotes although these differences were not significant.

Although we examined male and female populations separately like Tadić et al., associations with rs979605 only approached significance in these subpopulations. However, by analyzing males and females together, we observed a significant rs979605 × sex interaction. This finding suggests that genetic variants of *MAOA* may have an influence on the response to ATD therapy in depressed patients in a sex-dependent manner. Importantly, this observation could extend to other X-chromosomal genetic polymorphisms involved in ATD pharmacodynamics.

We also examined the plasma 5HIAA/5HT ratio as an estimate of MAO activity. Decreased platelet MAO activity was previously reported in female *MAOA* rs979605(G) carriers [5] and male *MAOB* rs1799836(T) carriers [27]. We observed a significant association of the plasma 5HIAA/5HT ratio with rs1799836 but not rs979605. Overall, TT/T females/males had a significantly higher ratio compared to CC/C females/males, which could be indicative of increased MAO activity, a result that contradicts previous findings [27]. However, compared to this study, we analyzed a population of depressed individuals. Importantly, MAO activity was shown to differ in depressed individuals compared to healthy controls [8,9]. MAO expression and activity is influenced by demographic—including age, sex, and smoking [5]—and biological factors, including epigenetics [37], which, at least for *MAOB*, is influenced by smoking [6]. We were able to control for the former but not the latter (at least not directly and not completely). Additionally, we estimated MAO activity from peripheral plasma 5HT and 5HIAA levels measured using ultraperformance liquid chromatography coupled with mass spectrometry (UPLC-MS), while the previous studies measured MAO activity using radiometric assays. Of note, most peripheral 5HT is stored in platelets that only contain MAOB, which only weakly metabolizes 5HT [7], while the rest is metabolized in peripheral organs [38]. Peripheral 5HT metabolism may differ from that of the brain. However, platelet MAO activity and MAOB density in the prefrontal cortex were both increased in depressed individuals [4,9], suggesting that MAO expression/activity in peripheral platelets may mirror that of the brain [4].

MAO expression is influenced by many factors. The sex hormones estrogen and progesterone can inhibit and promote 5HT degradation, respectively [39]. Glucocorticoids can also regulate brain MAOA activity [4,40]. *MAOA* and *MAOB* promoter regions contain binding elements for Sp1 [7], a transcription factor (TF) that regulates the expression of many genes. Altered Sp1 binding activity was observed in 3-week fluoxetine-treated rats [41], suggesting a role for Sp1 in the response to ATDs. At least for *MAOA*, members of the Sp family can bind to these Sp1-binding sites and either activate (Sp1, Sp4) or inhibit (Sp3) transcription [42]. Sp1 can also interact with transcription machinery and other TFs [42,43], including the glucocorticoid-regulated CDCA7L/R1 protein, and possibly the estrogen receptor [44]. The five *MAOA* genetic polymorphisms we identified were in LD. It is intriguing to speculate that some of these genetic polymorphisms are functional and may disrupt downstream processes linked to ATD therapy response, for example, through their interaction(s) with TFs. The potential and varied interactions between TFs and sex hormones in the context of MDD may contribute to the differential response to ATD treatment we observed between female rs979605(A) homozygotes and male A carriers.

Recent evidence suggests that pharmacogenomics-guided care improves treatment outcomes in depressed patients compared to treatment-as-usual, with 40% and 49% increases in response and remission, respectively [45]. It remains possible that an expanded genomic panel may further improve guided care. Polygenic risk scores also show promise as predictive tools although, to date, only nominal associations with antidepressant response in the context of MDD have been reported [46]. Given the differences between males and females in the context of depression [2], sex-specific factors may help improve the utility of each method. Although our results suggest a sex-dependent association of the *MAOA* rs979605 polymorphism with response following ATD treatment, this should be further replicated in an independent cohort of depressed individuals.

The present study has several limitations. First, the proportion of missing data was high. However, the proportion at M6 was comparable to that of the STAR*D cohort at 12 weeks [47]. Second, the genetic and metabolomic cohorts greatly differ in size. Studies with greater sample sizes and a balance of males and females would help corroborate our findings. Third, we only had access to peripheral metabolite levels, which may differ from those in the brain. This study benefits from its naturalistic and prospective design, which allows for the analysis of treatment response across a 6-month period in a “real-world” clinical setting. The modeling strategy by Chen et al. [48] also allowed us to more robustly analyze these X-chromosomal variants. To date, and to the best of our knowledge, this is the largest analysis of *MAOA*/*MAOB* variants in clinical response following ATD treatment in a predominantly Caucasian population of depressed patients.

## 4. Materials and Methods

### 4.1. Study Design

Do Antidepressants Induce Metabolic Syndromes (METADAP) is a 6-month prospective, multicentric, and observational cohort study carried out in a psychiatric setting [49]. Individuals with a current MDE were treated in naturalistic conditions and assessed before and during ATD treatment. This study was registered by the French National Agency for Medicine and Health Products Safety (ANSM) and the Commission Nationale de l’Informatique et des Libertés (CNIL). It was approved by the Ethics Committee of Paris-Boulogne (France) and conformed to international ethical standards (ClinicalTrials.gov identifier: NCT00526383).

### 4.2. Patient Population

Males and females 18–65 years of age without a serious medical or inflammatory condition were recruited. Inclusion criteria were presentation with a MDE in the context of MDD (DSM-IVTR)—assessed by the Mini International Neuropsychiatric Interview (MINI)—a score ≥ 18 on the HDRS [50], and need for a new ATD treatment. Individuals with psychotic disorder, bipolar disorder, alcohol or drug dependence, or an eating disorder were excluded, as previously described [49]. Measures and samples were obtained prior to beginning ATD treatment (M0) and at M1, M3, and M6.

Among the 643 individuals included in METADAP, 19 had major protocol deviations and were excluded. Of the 624 available for analysis, 519 individuals provided samples for genetic studies. From these, 400 underwent high-throughput sequencing. Nine samples were removed due to technical difficulties (*n* = 2), having a heterozygous call as a male (*n* = 1), or for having a missing genotype rate > 50% (*n* = 6). Ten samples without follow-up measures (i.e., M0 only) were removed given our objective to examine clinical response following ATD treatment. Three samples treated by MAO inhibitors were removed given our objective to examine the influence of *MAOA*/*MAOB* gene variants on clinical response, including their potential impact on MAO function. Thus, 378 individuals were analyzed (clinical cohort). Dropouts occurred mainly because of ATD changes, unauthorized drug use, or loss to follow-up. Ancestry was self-reported. Caucasian individuals were defined as having Caucasian parents; African individuals as having Sub-Saharan African and/or Afro-Caribbean parents; and Asian individuals as having East Asian, Central Asian, and/or South Asian parents [51]. In the original metabolite analysis of METADAP [38], 173 individuals were analyzed. Among them, 148 were present within the clinical cohort and analyzed (metabolomic cohort). All individuals provided written informed consent for study participation and genetic analyses [49].

### 4.3. Antidepressant Treatment

Monotherapies were prescribed by a psychiatrist in a “real world” psychiatric treatment setting as previously described [49] and belonged to one of five classes: SSRI; serotonin norepinephrine reuptake inhibitors (SNRI); tricyclic antidepressants (TCA); other ATDs; and electroconvulsive therapy (ECT). In this ancillary study of 378 individuals, 41% were prescribed an SSRI, 40% an SNRI, 7% a TCA, 9% another ATD treatment, and 3% received ECT. If a change in treatment was required during follow-up, the individual was dropped from the study.

### 4.4. Clinical Improvement after Antidepressant Treatment

The HDRS [50] was used to assess depression severity at M0 and clinical improvement after ATD treatment at M1, M3, and M6. Responders and remitters were defined by a decreased HDRS score ≥ 50% relative to baseline and a HDRS score ≤ 7 after ≥4 weeks of treatment, respectively. Clinical assessments were performed blind to genotyping results. Each interview and diagnostic assignment was reviewed by a senior psychiatrist. For each individual, all visits were reviewed by the same psychiatrist.

### 4.5. High-Throughput Sequencing, Sequence Alignment, and Variant Calling

For sequencing, 5 mL of whole blood was collected at baseline. Leukocytic DNA was extracted from 1 mL of blood using a Puregene Kit (Gentra systems, Minneapolis, MN, USA) and stored at −20 °C. DNA was sequenced using a targeted-exome panel of genes involved in mood disorders and ATD metabolism. The protocols for high-throughput sequencing [52] and information about the gene panel and variant calling [53] are described elsewhere.

### 4.6. Genetic Variant Selection

Variant call format data were loaded into R (v4.1.0) [54] using the vcfR package (v1.9.0) [55], as previously described [56]. Variant calls with a sequencing depth < 20, SNPs with a quality score (QUAL) < 275, insertion/deletions with a QUAL < 770, heterozygous calls with an allele balance (AB) < 0.34 or >0.79, and homozygous calls with an AB < 0.96 were annotated as poor-quality calls. Variants with a call rate (# poor-quality calls/# calls) < 95% were removed [57]. Variants with a MAF ≥ 5% were selected for further analysis.

### 4.7. Haplotype Analysis, Genotype Imputation, and Functional Annotation

HWE and LD were assessed using Haploview (v4.2) [58]. Variants were considered to be in LD by a *r*^2^ ≥ 0.8, as assessed in the subgroup of Caucasians comprising 91% of the study population. The variant with the highest MAF was selected as a haplotype proxy for further analysis.

### 4.8. 5HT and 5HIAA Measurement

Methods for metabolite level determination were previously described [38]. Metabolite extracts were prepared from 50 μL of fasting plasma stored at −80 °C via protein precipitation with 250 μL of acetonitrile, evaporation under nitrogen, and reconstitution in 50 μL of water. Three reference human plasma samples were prepared in each analytical batch as a control within and between analyses. UPLC-MS was performed using a Waters Acquity UPLC system coupled to a Thermo Scientific Q Exactive mass spectrometer. A proportion of values were below the lower limits of quantification (LLOQ) for 5HT (21.4%; LLOQ: 0.00113 μM) and 5HIAA (3.6%; 0.0131 μM) and were imputed as LLOQ/√2 (5HT:0.000799 μM; 5HIAA:0.00926 μM), as previously described [38].

### 4.9. Data Analysis

Statistical analyses were performed using R (v4.1.0) [54]. Quantitative variables were analyzed using nonparametric Wilcoxon rank sum or Kruskal–Wallis tests. Qualitative variables were analyzed using Fisher’s exact tests. Linear mixed-effects models and generalized linear mixed-effects models were constructed using the lme4 package (v1.1-27.1) [59]. The main variable to be explained was the HDRS total score. Other variables to explain were the response and remission rates and the plasma 5HIAA/5HT ratio. Chen et al. recently proposed a method (referred to as the proposed full model) that is robust to XCI, XCI skewness, and escape from XCI [48]. Briefly, unknown XCI status and XCI skewness were accounted for by analytical equivalencies to the variant × sex interaction and dominance term (i.e., female heterozygotes coded as 1 and all others as 0), respectively. Variant genotype and its interactions with time and sex were the main explanatory variables examined. Age, sex, the dominance term, time, and the sex × time interaction were included a priori as fixed-effects covariables in all full models. ATD class was also included a priori in all models to control for potential differences in treatment efficacy (in clinical models) or the plasma 5HIAA/5HT ratio (in metabolomic models) between treatment classes (e.g., SSRIs and ECT). As smoking and smoking history impact MAO activity [5,6], smoking status (i.e., non-smoker, current smoker, or former smoker) was also included a priori in all models. Since ATDs alter 5HT levels, ATD-naïve status was also included as a fixed-effect covariable in models of the plasma 5HIAA/5HT ratio. Sociodemographic and baseline clinical variables that significantly differed (i.e., *p* < 0.05) between genotypes were also included as fixed-effects covariables. Individual was included as a random-effect to account for repeated measures. Significance of fixed effects was assessed using the Satterthwaite method in linear mixed-effects models and the Wald chi-square test in generalized linear mixed-effects models. As one *MAOA* SNP and one *MAOB* SNP were analyzed, a Bonferroni-corrected threshold of *p* < 0.025 (0.05/2) was considered significant. If variant × sex interactions were significant in the full model, mixed-effects models were constructed in allelic subgroups under different XCI status assumptions (i.e., using different coding strategies). Post hoc comparisons were carried out using the emmeans package [60].

Power analyses were performed using the genpwr package assuming an additive genetic model [61]. MAFs of 29.8% (rs979605) and 45.6% (rs1799836) as observed in non-Finnish Europeans of gnomAD [36], an alpha level of 5%, and a power of 80% were used.

## 5. Conclusions

In conclusion, we observed that the *MAOA* rs979605(A>G) polymorphism was significantly associated with clinical improvement following ATD treatment in a sex-dependent manner. Female rs979605(A) homozygotes had higher HDRS scores compared to male A carriers after 6 months of treatment. Since rs979605 is linked to other *MAOA* genetic variants, this association may be due to the effect(s) of other linked variants. Given the role of MAOA in 5HT metabolism and previous associations of *MAOA* genetic variants with clinical improvement after ATD treatment, their potential as biomarkers for clinical improvement following ATD therapy should continue to be investigated.

## Figures and Tables

**Figure 1 ijms-24-00497-f001:**
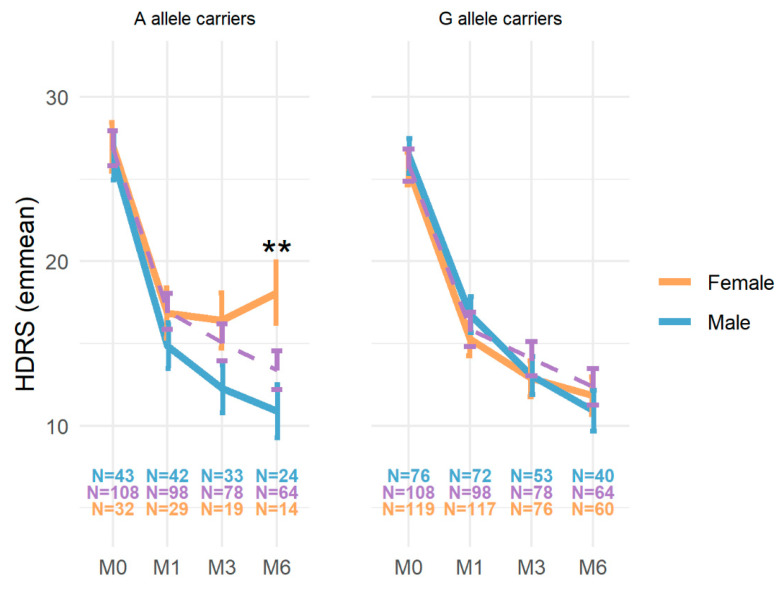
HDRS scores according to sex and *MAOA* rs979605 genotypes. Model estimates of the HDRS score (y-axis) at different study time points (x-axis) for *MAOA* rs979605 A allele female homozygotes (orange), female heterozygotes (purple dashed), and male carriers (blue). G allele carriers (right) are shown for comparison only. ** *p* < 0.01 following Bonferroni correction for the nine comparisons in A allele carriers. HDRS, 17-item Hamilton Depression Rating Scale.

**Figure 2 ijms-24-00497-f002:**
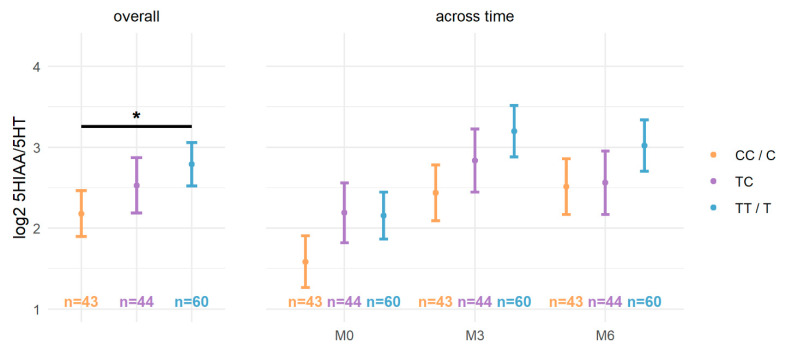
Log_2_ 5HIAA/5HT ratios according to *MAOB* rs1799836 genotypes. The log_2_ 5HIAA/5HT ratio (y-axis) according to rs1799836 genotypes (see legend) both overall (**left**) and across time (**right**). * *p* < 0.05 following Bonferroni correction for the three comparisons.

**Table 1 ijms-24-00497-t001:** Sociodemographic characteristics according to males and females.

		Total	Male	Female
Clinical cohort				
		*n* = 378	*n* = 119	*n* = 259
Age (in years) (m ± SD)		45.3 ± 13.3	45.9 ± 12.8	45 ± 13.6
Education (n(%))	Primary	35(9)	11(9)	24(9)
	High school	166(44)	45(38)	121(47)
	University	176(47)	62(52)	114(44)
Ethnicity (n(%))	Caucasian	343(91)	114(96)	229(88)
	African	24(6)	1(1)	23(9)
	Mixed	10(3)	4(3)	6(2)
Smoking status (n(%))	Current	144(38)	48(40)	96(37)
	Former	48(13)	22(18)	26(10)
	Non	186(49)	49(41)	137(53)
Pack years (m ± SD)		14 ± 13.5	14.4 ± 12	13.8 ± 14.3
Recurrent MDE (n(%))		278(74)	79(66)	199(77)
Onset age MDE (in years) (m ± SD)		35.2 ± 14.4	36.6 ± 15.1	34.5 ± 14
Baseline HDRS (m ± SD)		24.7 ± 4.9	24.8 ± 5	24.7 ± 4.8
Therapy (n(%))	SSRI	156(41)	49(41)	107(41)
	SNRI	152(40)	49(41)	103(40)
	TCA	24(7)	6(5)	18(7)
	Other	34(9)	9(8)	25(10)
	ECT	12(3)	6(5)	6(2)
Missing at follow-up (n(%))	M1	20(5)	5(4)	15(6)
	M3	119(31)	33(28)	86(33)
	M6	176(47)	55(46)	121(47)
Metabolomic cohort				
		*n* = 148	*n* = 52	*n* = 96
Age (in years) (m ± SD)		46.2 ± 12.2	45.6 ± 11.8	46.5 ± 12.4
Education (n(%))	Primary	15(10)	5(10)	10(10)
	High school	58(39)	15(29)	43(45)
	University	74(50)	31(60)	43(45)
Ethnicity (n(%))	Caucasian	139(94)	49(94)	90(94)
	African	7(5)	1(2)	6(6)
	Mixed	2(1)	2(4)	0(0)
Smoking status (n(%))	Current	53(36)	22(42)	31(32)
	Former	15(10)	9(17)	6(6)
	Non	80(54)	21(40)	59(61)
Pack years (m ± SD)		13.6(11.3)	14.6(11.9)	12.8(10.9)
Recurrent MDE (n(%))		100(68)	30(58)	70(73)
Onset age MDE (in years) (m ± SD)		36.7 ± 14.4	37.5 ± 15.7	36.2 ± 13.7
Baseline HDRS (m ± SD)		23.4 ± 4.1	23.6 ± 4.5	23.3 ± 3.9
ATD-naïve (n(%))		59(40)	22(42)	37(39)
Therapy (n(%))	SSRI	61(41)	20(38)	41(43)
	SNRI	61(41)	25(48)	36(38)
	TCA	12(8)	3(6)	9(9)
	Other	12(8)	4(8)	8(8)
	ECT	2(1)	0(0)	2(2)

Sociodemographic characteristics of the whole sample and in males and females of the clinical and metabolomic cohorts. ATD, antidepressant drug; ECT, electroconvulsive therapy; HDRS, 17-item Hamilton Depression Rating Scale; m, mean; M1, after 1 month of treatment; M3, after 3 months of treatment; M6, after 6 months of treatment; MDE, major depressive episode; n, number of individuals; SD, standard deviation; SNRI, serotonin norepinephrine reuptake inhibitor; SSRI, selective serotonin reuptake inhibitor; TCA, tricyclic antidepressant.

**Table 2 ijms-24-00497-t002:** Genomic and haplotypic information of the 6 selected *MAOA* and *MAOB* variants.

Rs#	Region	Variant Effect	RefSeq	Major Allele	Minor Allele	HWE_F_	MAF_Total_ (%)	MAF_F_ (%)	MAF_M_ (%)	*p*
*MAOA*										
rs6323	Exon 8	p.Arg297Arg	NM_000240	T	G	0.56	32.7	32.9	36.8	0.32
rs2235186	Intronic			G	A	0.56	32.9	32.9	38.2	0.32
rs979606	Intronic			T	C	0.51	32.8	32.8	37.8	0.32
rs979605	Intronic			G	A	0.78	34.8	33.6	38.2	0.62
rs1137070	Exon 14	p.Asp470Asp	NM_000240	C	T	0.63	34.0	33.8	36.8	0.59
				**Haplotype frequency**						
				65.6%	33.7%					
*MAOB*										
rs1799836	Intronic			T	C	0.99	45.1	45.8	38.7	0.08

For each variant, we show the rs identifier (rs#), its genomic region, its protein-level effect and the corresponding RefSeq accession, the major and minor alleles, HWE (in females), the total and female- and male-specific MAFs, and the *p*-value of the χ^2^ test comparing male and female frequencies. For *MAOA* variants, we also show the frequency of the major and minor allele haplotypes. F, female; HWE, Hardy–Weinberg equilibrium; M, male; MAF, minor allele frequency; *p*, *p*-value.

**Table 3 ijms-24-00497-t003:** Sociodemographic characteristics according to *MAOA* rs979605 genotypes in males and females.

		*MAOA* rs979605						
		Male			Female			
		G	A	*p*	GG	GA	AA	*p*
Clinical Cohort								
		*n* = 76	*n* = 43		*n* = 119	*n* = 108	*n* = 32	
Age (in years) (m ± SD)		46.8 ± 12.8	44.4 ± 12.9	0.28	43.3 ± 14.1	45.6 ± 13.5	49.3 ± 11	0.055
Education (n(%))	Primary	7(9)	4(9)	0.90	10(8)	10(9)	4(12)	0.24
	High school	27(36)	18(42)		48(40)	58(54)	15(47)	
	University	41(54)	21(49)		61(51)	40(37)	13(41)	
Ethnicity (n(%))	Caucasian	72(95)	42(98)	1.00	103(87)	99(92)	27(84)	0.46
	African	1(1)	0(0)		12(10)	8(7)	3(9)	
	Mixed	3(4)	1(2)		3(3)	1(1)	2(6)	
Smoking status (n(%))	Current	29(38)	19(44)	0.77	48(40)	37(34)	11(34)	0.52
	Former	14(18)	8(19)		14(12)	8(7)	4(12)	
	Non	33(43)	16(37)		57(48)	63(58)	17(53)	
Pack years (m ± SD)		14.1 ± 12	14.9 ± 12.3	0.92	13.6 ± 16.9	13.3 ± 12	16.6 ± 9.7	0.29
Recurrent MDE (n(%))		54(71)	25(58)	0.16	88(74)	87(81)	24(75)	0.46
Onset age MDE (in years) (m ± SD)		36.9 ± 14.2	36.1 ± 16.9	0.81	34.4 ± 14.8	33.9 ± 13	36.4 ± 14.4	0.70
Baseline HDRS (m ± SD)		25.3 ± 5.1	23.9 ± 4.8	0.13	24.5 ± 4.8	24.9 ± 5.1	24.9 ± 4.3	0.77
Therapy (n(%))	SSRI	30(39)	19(44)	0.81	49(41)	47(44)	11(34)	0.87
	SNRI	30(39)	19(44)		46(39)	43(40)	14(44)	
	TCA	4(5)	2(5)		8(7)	8(7)	2(6)	
	Other	7(9)	2(5)		13(11)	7(6)	5(16)	
	ECT	5(7)	1(2)		3(3)	3(3)	0(0)	
Missing at follow-up (n(%))	M1	4(5)	1(2)	0.65	2(2)	10(9)	3(9)	0.017 *
	M3	23(30)	10(23)	0.52	43(36)	30(28)	13(41)	0.26
	M6	36(47)	19(44)	0.85	59(50)	44(41)	18(56)	0.21
Metabolomic Cohort								
		*n* = 19	*n* = 33		*n* = 42	*n* = 42	*n* = 12	
Age (in years) (m ± SD)		45.3 ± 11	45.8 ± 12.5	0.91	44 ± 12.6	46.7 ± 12.4	54.2 ± 8.9	0.023 *
Education (n(%))	Primary	3(16)	2(6)	0.26	3(7)	6(14)	1(8)	0.69
	High school	3(16)	12(36)		17(40)	20(48)	6(50)	
	University	13(68)	18(55)		22(52)	16(38)	5(42)	
Ethnicity (n(%))	Caucasian	19(100)	30(91)	0.70	39(93)	39(93)	12(100)	1.00
	African	0(0)	1(3)		3(7)	3(7)	0(0)	
	Mixed	0(0)	2(6)		0(0)	0(0)	0(0)	
Smoking status (n(%))	Current	9(47)	13(39)	0.93	14(33)	13(31)	4(33)	0.97
	Former	3(16)	6(18)		3(7)	2(5)	1(8)	
	Non	7(37)	14(42)		25(60)	27(64)	7(58)	
Pack years (m ± SD)		15.8 ± 13.3	13.9 ± 11.4	0.70	7.5 ± 8.2	15.7 ± 11.5	19.8 ± 10.3	0.51
Recurrent MDE (n(%))		9(47)	21(64)	0.38	30(71)	31(74)	9(75)	1.00
Onset age MDE (in years) (m ± SD)		38.7 ± 16.8	36.8 ± 15.3	0.60	36.6 ± 14	35.1 ± 12.9	38.9 ± 16.2	0.80
Baseline HDRS (m ± SD)		22.5 ± 3.5	24.2 ± 4.9	0.30	23.6 ± 4.2	23.1 ± 3.8	23.4 ± 3.5	0.86
ATD-naïve (n(%))		6(32)	16(48)	0.46	14(33)	20(48)	3(25)	0.28
Therapy (n(%))	SSRI	6(32)	14(42)	0.79	14(33)	25(60)	2(17)	0.038 *
	SNRI	11(58)	14(42)		18(43)	13(31)	5(42)	
	TCA	1(5)	2(6)		5(12)	2(5)	2(17)	
	Other	1(5)	3(9)		3(7)	2(5)	3(25)	
	ECT	0(0)	0(0)		2(5)	0(0)	0(0)	

Sociodemographic characteristics of males and females of the clinical and metabolomic cohorts according to *MAOA* rs979605 genotypes. Males are hemizygous for either allele G or A. Females are either GG homozygotes, GA heterozygotes, or AA homozygotes. Wilcoxon rank-sum tests were used to compare age, pack years, onset age of MDE, and baseline HDRS (presented as mean ± standard deviation) between males, while Kruskal–Wallis tests were used to compare these variables between females. Fisher’s exact tests were used to compare education status, ethnicity, smoking status at baseline, MDE recurrence, prescribed therapy, missing at follow-up rates over time, and ATD-naïve status (presented as the group number and percentage). * *p* < 0.05. ATD, antidepressant drug; ECT, electroconvulsive therapy; HDRS, 17-item Hamilton Depression Rating Scale; m, mean; M1, after 1 month of treatment; M3, after 3 months of treatment; M6, after 6 months of treatment; MDE, major depressive episode; n, number of individuals; *p*, *p*-value; SD, standard deviation; SNRI, serotonin norepinephrine reuptake inhibitor; SSRI, selective serotonin reuptake inhibitor; TCA, tricyclic antidepressant.

**Table 4 ijms-24-00497-t004:** Sociodemographic characteristics according to *MAOB* rs1799836 genotypes in males and females.

		*MAOB* rs1799836						
		Male			Female			
		T	C	*p*	TT	TC	CC	*p*
Clinical Cohort								
		*n* = 71	*n* = 45		*n* = 70	*n* = 126	*n* = 61	
Age (in years) (m ± SD)		45.6 ± 12.7	46.8 ± 13.4	0.69	44.2 ± 13.7	45.4 ± 13.4	45.6 ± 14	0.85
Education (n(%))	Primary	7(10)	4(9)	1	5(7)	13(10)	6(10)	0.5
	High school	27(38)	17(38)		38(54)	52(41)	30(49)	
	University	36(51)	24(53)		27(39)	61(48)	25(41)	
Ethnicity (n(%))	Caucasian	68(96)	43(96)	0.57	66(94)	114(90)	47(77)	<0.001 *
	African	0(0)	1(2)		2(3)	7(6)	14(23)	
	Mixed	3(4)	1(2)		1(1)	5(4)	0(0)	
Smoking status (n(%))	Current	26(37)	21(47)	0.6	26(37)	43(34)	25(41)	0.19
	Former	14(20)	8(18)		10(14)	8(6)	8(13)	
	Non	31(44)	16(36)		34(49)	75(60)	28(46)	
Pack years (m ± SD)		16 ± 14.1	12.5 ± 9.3	0.73	12.5 ± 12.3	17 ± 17.3	10.2 ± 9.4	0.10
Recurrent MDE (n(%))		48(68)	30(67)	1	53(76)	94(75)	50(82)	0.54
Onset age MDE (in years) (m ± SD)		36.6 ± 15.1	36.4 ± 15.7	0.91	35.9 ± 13.3	34.2 ± 14.4	34 ± 14.1	0.60
Baseline HDRS (m ± SD)		24.2 ± 4.6	25.5 ± 5.6	0.23	24.3 ± 4.9	25.1 ± 4.7	24.4 ± 5.1	0.31
Therapy (n(%))	SSRI	31(44)	17(38)	0.24	27(39)	54(43)	25(41)	0.37
	SNRI	26(37)	22(49)		30(43)	52(41)	20(33)	
	TCA	6(8)	0(0)		5(7)	10(8)	3(5)	
	Other	5(7)	3(7)		7(10)	7(6)	11(18)	
	ECT	3(4)	3(7)		1(1)	3(2)	2(3)	
Missing at follow-up (n(%))	M1	2(3)	2(4)	0.64	3(4)	9(7)	3(5)	0.78
	M3	16(23)	16(36)	0.14	28(40)	36(29)	21(34)	0.25
	M6	23(51)	23(51)	0.44	35(50)	56(44)	29(48)	0.75
Metabolomic Cohort								
		*n* = 33	*n* = 19		*n* = 27	*n* = 44	*n* = 24	
Age (in years) (m ± SD)		43.5 ± 12	49.2 ± 11	0.91	44.3 ± 13	48.3 ± 12.6	46.5 ± 10.7	0.50
Education (n(%))	Primary	3(9)	2(11)	0.26	3(11)	5(11)	2(8)	0.050
	High school	10(30)	5(26)		11(41)	15(34)	17(71)	
	University	19(58)	12(63)		13(48)	24(55)	5(21)	
Ethnicity (n(%))	Caucasian	31(94)	18(95)	0.70	27(100)	41(93)	21(88)	0.16
	African	0(0)	1(5)		0(0)	3(7)	3(12)	
	Mixed	2(6)	0(0)		0(0)	0(0)	0(0)	
Smoking status (n(%))	Current	14(42)	8(42)	0.93	8(30)	11(25)	11(46)	0.018 *
	Former	5(15)	4(21)		3(11)	0(0)	3(12)	
	Non	14(42)	7(37)		16(59)	33(75)	10(42)	
Pack years (m ± SD)		14.5 ± 13.4	14.8 ± 10.1	0.70	10.5 ± 10.6	17.7 ± 12.2	11.8 ± 9.8	0.054
Recurrent MDE (n(%))		20(61)	10(53)	0.38	19(70)	30(68)	20(83)	0.39
Onset age MDE (in years) (m ± SD)		36 ± 15.7	40.2 ± 15.8	0.60	36.9 ± 12.7	37 ± 14.1	34.8 ± 14.5	0.77
Baseline HDRS (m ± SD)		23.5 ± 4.3	23.8 ± 4.9	0.30	22 ± 3.2	23.7 ± 3.8	23.9 ± 4.5	0.15
ATD-naïve (n(%))		12(36)	10(53)	0.46	11(41)	18(41)	8(33)	0.70
Therapy (n(%))	SSRI	13(39)	7(37)	0.79	9(33)	20(45)	12(50)	0.14
	SNRI	14(42)	11(58)		13(48)	15(34)	7(29)	
	TCA	3(9)	0(0)		2(7)	7(16)	0(0)	
	Other	3(9)	1(5)		2(7)	2(5)	4(17)	
	ECT	0(0)	0(0)		1(4)	0(0)	1(4)	

Sociodemographic characteristics of males and females of the clinical and metabolomic cohorts according to *MAOB* rs1799836 genotypes. Males are hemizygous for either allele T or C. Females are either TT homozygotes, TC heterozygotes, or CC homozygotes. Wilcoxon rank-sum tests were used to compare age, pack years, onset age of MDE, and baseline HDRS (presented as mean ± standard deviation) between males, while Kruskal–Wallis tests were used to compare these variables between females. Fisher’s exact tests were used to compare education status, ethnicity, smoking status at baseline, MDE recurrence, prescribed therapy, missing at follow-up rates over time, and ATD-naïve status (presented as the group number and percentage). * *p* < 0.05. ATD, antidepressant drug; ECT, electroconvulsive therapy; HDRS, 17-item Hamilton Depression Rating Scale; m, mean; M1, after 1 month of treatment; M3, after 3 months of treatment; M6, after 6 months of treatment; MDE, major depressive episode; n, number of individuals; *p*, *p*-value; SD, standard deviation; SNRI, serotonin norepinephrine reuptake inhibitor; SSRI, selective serotonin reuptake inhibitor; TCA, tricyclic antidepressant.

**Table 5 ijms-24-00497-t005:** Clinical associations with *MAOA* rs979605 and *MAOB* rs1799836 genotypes using the proposed full model for X-chromosomal variants.

	HDRS						Response			Remission		
	SS	MS	df (num)	df (den)	*F*	*p*	χ^2^	df	*p*	χ^2^	df	*p*
*MAOA* rs979605												
Age	0.44	0.44	1	369.04	0.015	0.90	2.68	1	0.10	0.78	1	0.38
Sex	149.07	149.07	1	385.70	4.97	0.026	1.44	1	0.23	2.84	1	0.092
ATD class	122.64	30.66	4	374.77	1.02	0.40	2.92	4	0.57	6.22	4	0.18
Loss to follow-up (M1)	86.28	86.28	1	442.40	2.88	0.091	1.56	1	0.21	0.32	1	0.57
Smoking status	19.39	9.70	2	369.28	0.32	0.72	3.70	2	0.16	0.40	2	0.82
Time	28,305.80	9435.27	3	857.69	314.42	<0.001 *	35.66	2	<0.001 *	36.81	2	<0.001 *
rs979605 dominance term	7.16	7.16	1	894.03	0.24	0.63	0.011	1	0.91	1.02	1	0.31
rs979605	0.14	0.14	1	392.04	0.0048	0.94	0.12	1	0.73	0.025	1	0.88
Time:rs979605	181.21	30.20	6	867.34	1.01	0.42	1.77	4	0.78	8.04	4	0.090
Sex:rs979605	189.65	189.65	1	367.50	6.32	0.012 *	2.81	1	0.094	3.27	1	0.071
Sex:Time	251.64	83.88	3	868.36	2.80	0.039	3.00	2	0.22	0.22	2	0.90
*MAOB* rs1799836												
Age	5.93	5.93	1	359.18	0.20	0.66	2.77	1	0.10	0.97	1	0.32
Sex	4.12	4.12	1	352.08	0.14	0.71	0.99	1	0.32	2.11	1	0.15
ATD class	156.33	39.08	4	368.79	1.29	0.27	3.69	4	0.45	6.46	4	0.17
Ethnicity	81.42	40.71	2	366.62	1.35	0.26	4.35	2	0.11	0.29	2	0.86
Smoking status	16.58	8.29	2	364.20	0.27	0.76	3.10	2	0.21	0.10	2	0.95
Time	35,802.44	11,934.15	3	853.23	395.00	<0.001 *	32.94	2	<0.001 *	39.08	2	<0.001 *
rs1799836 dominance term	31.48	31.48	1	845.36	1.04	0.31	0.95	1	0.33	1.47	1	0.22
rs1799836	119.36	119.36	1	356.46	3.95	0.048	1.52	1	0.22	2.17	1	0.14
Time:rs1799836	121.70	20.28	6	856.73	0.67	0.67	4.45	4	0.35	1.85	4	0.76
Sex:rs1799836	3.34	3.34	1	354.56	0.11	0.74	0.74	1	0.39	0.26	1	0.61
Sex:Time	138.63	46.21	3	855.42	1.53	0.21	4.23	2	0.12	1.11	2	0.57

Results of mixed-effects models of the HDRS score, response, and remission using the proposed full model. HDRS was assessed using the Satterthwaite method while response and remission were assessed using the Wald chi-square method. * *p* < 0.025. ATD, antidepressant drug; den, denominator; df, degrees of freedom; *F*, *F* statistic; HDRS, 17-item Hamilton Depression Rating Scale; MS, mean squares; num, numerator; *p*, *p*-value; SS, sum of squares; χ^2^, chi-square.

**Table 6 ijms-24-00497-t006:** Plasma 5HIAA/5HT ratio associations with *MAOA* rs979605 and *MAOB* rs1799836 genotypes.

	SS	MS	df (num)	df (den)	*F*	*p*
*MAOA* rs979605						
Age	11.61	11.61	1	135.08	5.99	0.016 *
Sex	0.33	0.33	1	136.39	0.17	0.68
ATD class	46.99	11.75	4	136.21	6.07	<0.001 *
Smoking status	2.66	1.33	2	136.27	0.69	0.50
ATD-naïve status	22.77	22.77	1	376.32	11.76	<0.001 *
Time	41.03	20.52	2	300.99	10.60	<0.001 *
rs979605 dominance term	0.02	0.02	1	330.21	0.0083	0.93
rs979605	2.55	2.55	1	135.88	1.32	0.25
Time:rs979605	0.26	0.07	4	285.41	0.034	1.00
Sex:rs979605	3.45	3.45	1	135.91	1.78	0.18
Sex:Time	3.81	1.91	2	285.33	0.98	0.38
*MAOB* rs1799836						
Age	15.53	15.53	1	133.66	8.02	<0.001 *
Sex	0.71	0.71	1	134.63	0.37	0.55
ATD class	42.30	10.58	4	134.71	5.47	<0.001 *
Smoking status	2.47	1.23	2	134.54	0.64	0.53
ATD-naïve status	23.74	23.74	1	372.91	12.27	<0.001 *
Time	40.46	20.23	2	298.70	10.45	<0.001 *
rs1799836 dominance term	0.00	0.00	1	313.20	0.0023	0.96
rs1799836	11.46	11.46	1	133.94	5.92	0.016 *
Time:rs1799836	2.91	0.73	4	282.18	0.38	0.83
Sex:rs1799836	0.25	0.25	1	134.42	0.13	0.72
Sex:Time	2.42	1.21	2	282.57	0.63	0.54

Results for mixed-effects models of the 5HIAA/5HT ratio assessed using the Satterthwaite method. * *p* < 0.025. ATD, antidepressant drug; den, denominator; df, degrees of freedom; *F*, *F* statistic; MS, mean squares; num, numerator; *p*, *p*-value; SS, sum of squares.

## Data Availability

The data presented in this study data are under the protection of health data regulation set by the French National Commission on Informatics and Liberty (Commission Nationale de l’Informatique et des Libertés, CNIL) in line with European regulations, the Data Protection Act, and the Comité de protection des personnes (CPP, equivalent to the Research Ethics Committee). Data are not publicly available, as French law forbids free access to METADAP data. Data are available upon reasonable request to the principal investigator of the study (emmanuelle.corruble@aphp.fr).

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
