# Peer review of "The MAOA rs979605 Genetic Polymorphism Is Differentially Associated with Clinical Improvement Following Antidepressant Treatment between Male and Female Depressed Patients"

_ijms, 2022, doi:10.3390/ijms24010497_

Round 1

Reviewer 1 Report

Dear Colleagues,

I have read your paper with great interest. In this study, MAOA rs979605 polymorphism was significantly associated with a AD clinical improvement. This study suggest that this variant "should be" considered as biomarker for clinical improvemen.

There are several comments:

1. Section 2.4 (line 176), the HDRS has evaluate at M0, M1, M3 and M6. But it is not clear if it has been analyzed according to each type of drug. What treatment has the best clinical improvement?

2. Section 2.5 (line 184), there is a lack of information on how the biological sample for genetic analysis was obtained, what type of biological sample was taken, and at what point in the study.

3. Section 2.8 (lines 204). "2.85. HT and 5HIAA measurement" Typographic error. What is the normal reference rate of the serotonin and its metabolite ratio? Does it vary depending on the type of drug? The mechanism of action of the drug can vary serotonin levels?

4. Table 1 is very extensive and it is difficult to pay attention to the information it indicates. I recommend reducing it to make it more visual and with the most relevant values.

5. Line 147: writting error. Don't start a sentence with a number, start with a letter. Lines 150, 152 and 154 are correct.

6. The study is very complete at the genetic level, but I lack clinical applicability in the discussion. Pharmacogenetics in antidepressants is controversial, especially regarding pharmacodynamics. If one patient not respond adequately to AD, this genetic testing might be considered?. And here the biggest doubt arises, for which drug could I apply the genetic test?

Reviewer 2 Report

Thanks for the opportunity to review this interesting paper. This study aims to  analyze the association of MAOA and MAOB genetic 35 variants with 1) clinical improvement and 2) the plasma 5HIAA/5HT ratio in 6-month ATD-treated depressed individuals. I have only several comments on this study. The study concluded that the MAOA rs979605 polymorphism, associated with the HDRS score in a sex-dependent manner, could be a useful biomarker for the response to ATD treatment. I have only a few comments.

1. Introduction: Please provide a through review of the research topic and state what value can this study bring to the current research evidence.

2. Results: Table 1 is not very clearly presented, can it be separated?

3. Discussion: Please do not repeat the results, instead highlight the results, and compared the results with previous studies and explain the results, and underscore the implications for research and practice.
